# Ctrl-U: Robust Conditional Image Generation via Uncertainty-aware Reward Modeling

**Guiyu Zhang**[*1,2]    **Huan-ang Gao**[*2]    **Zijian Jiang**[2]    **Hao Zhao**[†2]    **Zhedong Zheng**[†1]

[1] FST and ICI, University of Macau [2] AIR, Tsinghua University

## Abstract

In this paper, we focus on the task of conditional image generation, where an image is synthesized according to user instructions. The critical challenge underpinning this task is ensuring both the fidelity of the generated images and their semantic alignment with the provided conditions. To tackle this issue, previous studies have employed supervised perceptual losses derived from pre-trained models, *i.e.*, reward models, to enforce alignment between the condition and the generated result. However, we observe one inherent shortcoming: considering the diversity of synthesized images, the reward model usually provides inaccurate feedback when encountering newly generated data, which can undermine the training process. To address this limitation, we propose an uncertainty-aware reward modeling, called **Ctrl-U**, including uncertainty estimation and uncertainty-aware regularization, designed to reduce the adverse effects of imprecise feedback from the reward model. Given the inherent cognitive uncertainty within reward models, even images generated under identical conditions often result in a relatively large discrepancy in reward loss. Inspired by the observation, we explicitly leverage such prediction variance as an uncertainty indicator. Based on the uncertainty estimation, we regularize the model training by adaptively rectifying the reward. In particular, rewards with lower uncertainty receive higher loss weights, while those with higher uncertainty are given reduced weights to allow for larger variability. The proposed uncertainty regularization facilitates reward fine-tuning through consistency construction. Extensive experiments validate the effectiveness of our methodology in improving the controllability and generation quality, as well as its scalability across diverse conditional scenarios, including segmentation mask, edge, and depth conditions. Codes are publicly available at https://grenoble-zhang.github.io/Ctrl-U.

## 1 Introduction

Driven by the emergence of large-scale image-text datasets (Schuhmann et al., 2021; Changpinyo et al., 2021; Schuhmann et al., 2022) and the development of diffusion models (Dhariwal & Nichol, 2021; Nichol et al., 2021; Rombach et al., 2022), text-to-image (T2I) diffusion models (Ramesh et al., 2021; Saharia et al., 2022) have become fundamental in the field of controllable visual generation. These models excel at producing realistic, high-quality images that accurately reflect the descriptions provided in natural language. However, due to their inherent properties, text-based conditions often struggle to convey the detailed controls required for final generation results efficiently. This limitation becomes especially evident in certain scenarios where text prompts alone can not capture all necessary details, such as when depicting unique artistic styles or accurately rendering complex scenes. To this end, alongside text descriptions, a substantial body of research focuses on integrating novel conditional controls, such as user-drawn sketches and semantic masks, into T2I diffusion models (Qin et al., 2023; Ye et al., 2023; Zhang et al., 2023a; Mou et al., 2024). Despite investigations into the controllability of T2I diffusion models and their expanded applications (Gao et al., 2024; Li et al., 2024b; Xu et al., 2024; Gu et al., 2024), achieving precise and fine-grained control remains challenging. The primary issue lies in ensuring both the fidelity of the generated

---

[*]Equal contribution. The work is done during G Zhang's internship at the University of Macau.
[†]Corresponding authors.

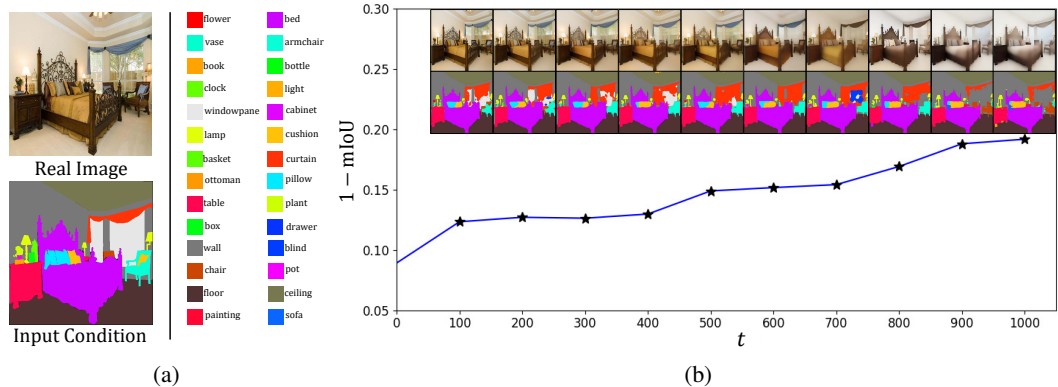

(a)  (b)

Figure 1: Given a test image and the layout condition, we employ a diffusion model to generate new images by adding noise and then recovering from the noisy input. **(a)** Ground-truth segmentation results with the category illustration. **(b)** Here we show the reward changes, *i.e.*, mIoU error, on newly generated images at different timesteps. The horizontal axis represents the current timestep $t$ and the vertical axis shows the error, *i.e.*, 1-mIoU. As shown, even at $t = 0$, there are non-zero mIoU errors. As $t$ increases, **although the visual layout aligns with the condition, the reward model tends to increase the error,** leading to the backpropagation of incorrect gradients.

images and their semantic alignment with the provided conditions. To tackle this issue, some efforts, *e.g.*, ControlNet++ (Li et al., 2024a) attempt to employ a pre-trained reward model to extract the corresponding condition from the generated images and enforce alignment between the specified condition and the output. However, we observe that the reward model inevitably produces inaccurate feedback (see Fig. 1). During the diffusion training, we add different levels of Guassian noise to the input, which increases the reconstruction challenge. It also leads to the distribution discrepancy between the generated image and the real image. As the timestep $t$ increases, such generation discrepancy increases. The reward model has not "seen" such generation discrepancy before, resulting false segmentation prediction, even if the generation is correctly aligned with the condition. Due to the diffusion compression process, we observe that even if $t = 0$, there is some mis-alignment feedback as well. If we do not rectify such misleading rewards, enforcing alignment between the provided conditions and the inaccurate predictions on newly generated data will consistently compromise the training process.

To mitigate the adverse effects of inaccurate rewards, we introduce a robust, controllable image generation approach via uncertainty-aware reward modeling (**Ctrl-U**). Motivated by our observation, we explicitly model the reward uncertainty to facilitate the reward back-propagation. The proposed uncertainty-based method consists of two phases, *i.e.*, uncertainty estimation and uncertainty regularization. **(1)** In the uncertainty estimation phase, we forward the identical input condition twice with different noise timesteps, yielding two similar generation results. We explicitly leverage the reward variance between these two generations as an uncertainty indicator. Notably, we do not introduce extra parameters and thus do not impact the inference efficiency of the diffusion model. **(2)** Following the uncertainty estimation, we adaptively adjust the loss weights of different reward feedback. Specially, the alignment reward for each pixel are rectified by its corresponding pixel-wise uncertainty. Rewards with lower uncertainty, indicating greater stability, should be given higher weights to encourage the model to learn from these reliable signals. Conversely, rewards with higher uncertainty, which are less stable, should be assigned reduced weights to minimize the negative impact of potentially inaccurate feedback. Quantitative and qualitative experiments verify the efficacy of the proposed method on controllability and image quality across various conditions.

- We observe an inherent drawback in enforcing alignment for conditional image generation using a pre-trained reward model, as the reward model often fails to generalize to newly generated data. To address the adverse effects of inaccurate reward feedback on conditional image generation, we introduce an uncertainty-aware reward modeling approach, termed Ctrl-U, which adaptively regularizes the reward learning process.

- Through extensive experiments on five benchmarks across three datasets, *i.e.*, ADE20k, COCO-Stuff, and MultiGen-20M, we validate the effectiveness of our methodology in improving controllability and generation quality. Our approach also shows scalability across various conditional scenarios, including segmentation masks, edges, and depth conditions.

## 2 RELATED WORKS

**Conditional Generation.** Recently, diffusion probabilistic models (Ho et al., 2020; Song et al., 2020a) have become an important cornerstone in general image generation. As the quality of generated images improves, a critical challenge remains, *i.e.*, achieving precise control over these generative models to meet the intricate and varied demands of real-world applications adequately. Motivated by the development of guidance mechanism (Song et al., 2020b; Ho & Salimans, 2022), T2I diffusion models such as GLIDE (Nichol et al., 2021), Imagen (Saharia et al., 2022), DALL·E 2 (Ramesh et al., 2022) excel at modeling fine-grained structures and texture details. To facilitate the training of diffusion models on limited resources, Latent Diffusion Model (LDM) (Rombach et al., 2022) maps the diffusion process from pixel space to the latent space. Building upon the foundation of LDM framework, Stable Diffusion (SD) exhibits exceptional capabilities in T2I generation, gaining widespread usage within the community due to its open-source models. Despite the astonishing capabilities of T2I diffusion models, language, with its sparse and high-level semantic nature, is unsuitable for processing intricate and low-level control images, *e.g.*, depth maps. To achieve conditional control in T2I diffusion models, researchers are exploring the integration of various control signals with text descriptions. ControlNet (Zhang et al., 2023a) integrates image-based conditions by incorporating an additional encoder copy into frozen T2I diffusion models via zero convolutions. This additional module links to the original UNet layers facilitates the integration of conditional inputs and prevent degradation in performance. Similarly, GLIGEN (Li et al., 2023) and T2I-Adapter (Mou et al., 2024) propose utilizing independent adapters (or extra modules) to synchronize internal knowledge within T2I models with external conditions. Furthermore, Cocktail (Hu et al., 2023) integrates multi-modal control information to yield more high-quality outputs. With the rapid development of large language models, some recent works have explored prompt engineering for regulated generation. For instance, ReCo (Zhang et al., 2023b) specifies text descriptions of different regions to model the fine-grained structure and aesthetic features. Control-GPT (Zhang et al., 2023b) applies GPT-4 (Achiam et al., 2023) as the sketch generator to incorporate control signals. However, one drawback is that their generated images usually deviate from input conditions. In this work, we take a closer look at denoised results and explicitly introduce the uncertainty to impose consistency constraints adaptively.

**Uncertainty-aware learning.** As data-driven technologies rapidly advance, the imperative for model reliability grows. Measuring the "confidence" of predictions has been a long-standing problem. As a result, researchers are progressively focusing on uncertainty as a viable solution. Kendall & Gal (2017) categorize uncertainty into two primary types: epistemic uncertainty and aleatoric uncertainty. The former epistemic uncertainty denotes model uncertainty, reflecting variations in model weights even when trained on the same dataset. The representative approach in this direction is Bayesian networks (Jensen & Nielsen, 2007; Neal, 2012), which focuses on learning the distribution of weights and estimates uncertainty by assessing the distribution variance. Similarly, Monte Carlo Dropout (Gal & Ghahramani, 2016) is proposed to randomly use varying dropout masks to simulate variational Bayesian approximation effectively. Another line of research on epistemic uncertainty (Raghu et al., 2019; Nandy et al., 2020; Lee & AlRegib, 2020) adapts the original model to measure prediction uncertainty directly. However, incorporating this additional uncertainty estimation typically compromises both the training efficiency and predictive accuracy of the model. In addition, ensemble methods (Lakshminarayanan et al., 2017; Malinin et al., 2019; Wenzel et al., 2020) combine various deterministic models in the prediction process to improve prediction accuracy, but is constrained by the computational burden associated with operating multiple independent networks and the requisite diversity across ensemble models. Another family of research tackles the inherent uncertainty in data, arising from noise interference or ambiguous annotations. This methodology for addressing uncertainty has been applied across various fields: such as person re-identification (Zhang et al., 2022), image retrieval (Chen et al., 2022), image classification (Litrico et al., 2023), hallucination detection (Zhang et al., 2024b) and 3D object detection (Zhang et al., 2024a). In particular, Dou et al. (2022) effectively simulates uncertainty by utilizing model prediction variance to inject noise into the latent space. In a manner akin to (Kendall & Gal, 2017), a dynamic uncertainty-based loss is introduced to enhance training stability. Unlike existing uncertainty-based works, we explicitly introduce noise at varying intensities into the diffusion model to enable the quantification of uncertainty. It is worth noting that we explore the utilization of uncertainty in the untapped conditional image generation task, preserving the original diffusion learning and mitigating the adverse effects of inaccurate feedback from the reward model.

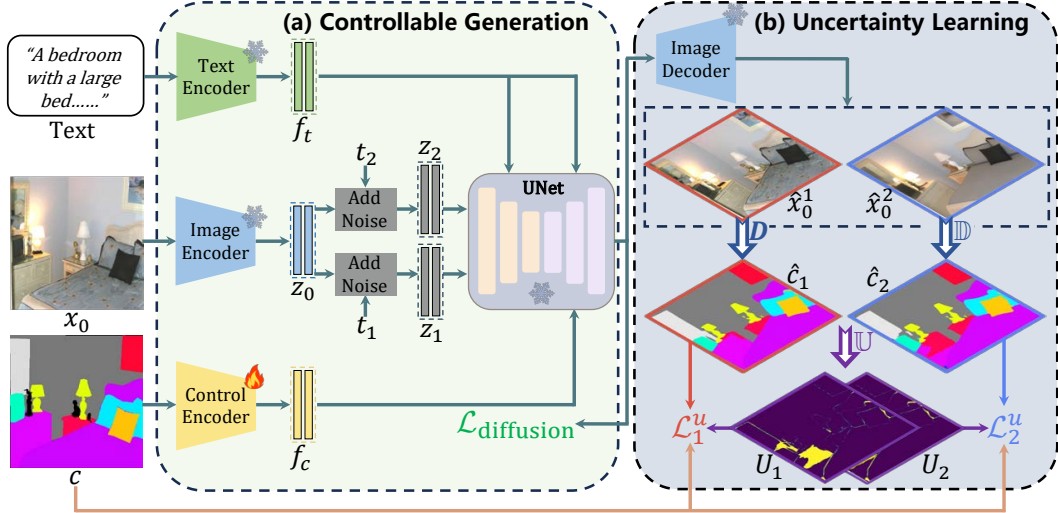

Figure 2: **A brief overview of our pipeline.** Here, we take the segmentation mask as a conditional generation example. **(a)** Conditional Generation. Given text, source image $x_0$, and the conditional control $c$, we extract feature $z_0$, $f_t$, $f_c$, respectively. Then, we fine-tune the Diffusion model to generate two intermediate features for the image decoder. **(b)** Uncertainty Learning. Given the two features, we decode the two images, *i.e.*, $\hat{x}_0^1$ and $\hat{x}_0^2$. Then we apply the reward model to obtain the two layout predictions $\hat{c}_1$ and $\hat{c}_2$. We leverage the KL-divergence prediction discrepancy between $\hat{c}_1$, $\hat{c}_2$ as the uncertainty indicator $U_1$, $U_2$ (see Eq. 2). Based on $U_1$, $U_2$, we then calculate the rectified reward loss between the predicted label $\hat{c}_1$, $\hat{c}_2$ and the ground-truth label $c$, as Eq. 3.

## 3 METHOD

### 3.1 UNCERTAINTY ESTIMATION

We provide a brief overview of the pipeline in Fig. 2. **To simplify the explanation, here we take the segmentation task as an example, if not specified.** Given the triplet input, *i.e.*, one source image input $x_0$, one text prompt, and the conditional control $c$, the basic encoders extract the visual feature $z_0$ of $x_0$, textual feature $f_t$ and the control feature $f_c$ of $c$. We fix the weight of the off-the-shelf image encoder and the text encoder, while finetuning the control encoder. During the conditional diffusion training, we first add Gaussian noise $\epsilon$ to the feature map $z_0$ as the noisy latent. In particular, we conduct two generation forward with identical condition $c$ but different $t_1$ and $t_2$, and resampled Gaussian noise $\epsilon$. Therefore, the two noisy latents $z_1$ and $z_2$ can be formulated as:

$$z_1 = \sqrt{\bar{\alpha}_{t_1}} z_0 + \sqrt{1 - \bar{\alpha}_{t_1}} \epsilon, \quad z_2 = \sqrt{\bar{\alpha}_{t_2}} z_0 + \sqrt{1 - \bar{\alpha}_{t_2}} \epsilon, \quad \epsilon \sim \mathcal{N}(\mathbf{0}, \mathbf{I}), \quad (1)$$

Following the ControlNet pipeline (Zhang et al., 2023b), we further fuse text condition $f_t$ and image condition $f_c$ to predict the added noise. After removing the predicted noise, we obtain the recovered latent, *i.e.*, $\hat{z}_0^1$ and $\hat{z}_0^2$. Then, given the latent $\hat{z}_0^1$ and $\hat{z}_0^2$, the pretrained decoder is to reconstruct the input image as $\hat{x}_0^1$ and $\hat{x}_0^2$, respectively. To align the condition within the generated images, we follow the existing works and apply an off-the-shelf reward model $D$ to quantify the consistency between the input condition $c$ and the corresponding output condition of the generated images. However, as shown in Fig. 1b, the reward model usually contains inaccurate feedback, even if the visual layout has already aligned with the condition. If we consistently apply strong supervision on the consistency between inaccurate rewards and conditions predicted on the newly generated data, the backpropagation of incorrect gradients will significantly compromise the model. To estimate inaccurate rewards, we explicitly leverage two diffusion forwards for the identical input conditions. We compare the reward discrepancy between extracted conditions $\hat{c}_1$, $\hat{c}_2$ from generated images, which can be considered as a reward indicator at the current timestep. For the segmentation map condition, considering the probability output, we quantify uncertainty by calculating the KL-divergence between the extracted conditions of two generations at the pixel level as:

$$U_1 = \mathbb{E}\left[\hat{c}_1 \log\left(\frac{\hat{c}_1}{\hat{c}_2}\right)\right] = \mathbb{E}\left[D(\hat{x}_0^1) \log\left(\frac{D(\hat{x}_0^1)}{D(\hat{x}_0^2)}\right)\right],$$

$$U_2 = \mathbb{E}\left[\hat{c}_2 \log\left(\frac{\hat{c}_2}{\hat{c}_1}\right)\right] = \mathbb{E}\left[D(\hat{x}_0^2) \log\left(\frac{D(\hat{x}_0^2)}{D(\hat{x}_0^1)}\right)\right]. \quad (2)$$

For other non-probability conditions, *e.g.*, edge and depth, we adopt the $\ell_1$ distance $U_1 = U_2 = |\hat{c}_1 - \hat{c}_2|$ as the uncertainty indicator.

**Discussion. 1). Why not use an auxiliary network to directly regress uncertainty?** Some existing methods (Zheng & Yang, 2021; Jain et al., 2024) introduce extra modules to model the uncertainty. However, directly regressing uncertainty often leads to overfitting, where the model predicts zero uncertainty for all samples or a large, uniform value. This line of approaches is challenging to optimize effectively. Therefore, instead of directly regressing uncertainty, we perform predictions twice to estimate the prediction variance, which serves as a cognitive uncertainty indicator for the reward model. Additionally, this method has the side benefit of not introducing the extra training parameters. **2). How about choosing the same timestep for the two-time generations?** Using the same timestep for the two-time generations is possible, and the samples will still differ due to the inherent randomness in $\epsilon$. However, this approach limits the diversity of the generated images, which hinders the full utilization of the reward model. In our experiments, we observe that using different timesteps for the two generated images helps to learn a better and more robust generation model after rectifying the reward.

## 3.2 UNCERTAINTY REGULARIZATION

The existing controllability modeling methods (Li et al., 2024a) usually adopt consistency loss between the input condition and the extracted condition, which can be regarded as a pixel-wise supervision. For example, when using segmentation mask as input condition, $\mathcal{L}^c$ can be defined as a per-pixel cross-entropy loss as:

$$\mathcal{L}_1^c = -c\log(\hat{c}_1), \quad \mathcal{L}_2^c = -c\log(\hat{c}_2) \tag{3}$$

where $c$ denotes the input condition, $\hat{c}_1, \hat{c}_2$ represent extracted conditions from generated images. Our objective is to adaptively rectify the inaccurate reward feedback. To achieve this, we adapt the original consistency loss by incorporating our estimated uncertainty:

$$\mathcal{L}_1^u = \frac{\mathcal{L}_1^c}{\exp(U_1)} + \lambda \cdot U_1, \quad \mathcal{L}_2^u = \frac{\mathcal{L}_2^c}{\exp(U_2)} + \lambda \cdot U_2, \tag{4}$$

where $\lambda$ denotes the regularization factor. The second term $\lambda \cdot U$ is to prevent the model from consistently predicting high uncertainty across all samples. The first term diminishes the impact of reward feedback when significant disagreement is evident, yet retains its influence when predictions are consistent. Notably, when the uncertainty value is constant, the back-propagation gradient is identical to the original consistency loss. Finally, to optimize robust conditional image generation, we adopt a combination of the diffusion loss $\mathcal{L}_{\text{diffusion}}$ and the proposed uncertainty-regularization loss $\mathcal{L}_1^u, \mathcal{L}_2^u$. The original diffusion loss at timestep $t_1$ and $t_2$ can be reformulated as:

$$\mathcal{L}_{\text{diffusion}} = \mathbb{E}\left[\|\epsilon_\theta(z_1, t_1, f_t, f_c) - \epsilon\|_2^2 + \|\epsilon_\theta(z_2, t_2, f_t, f_c) - \epsilon\|_2^2\right]. \tag{5}$$

Therefore, the total loss is as follows:

$$\mathcal{L}_{\text{total}} = \mathcal{L}_{\text{diffusion}} + \mu \cdot (\mathcal{L}_1^u + \mathcal{L}_2^u), \tag{6}$$

During training, we balance the ratio $\mu$ of diffusion training and reward feedback by:

$$\mu = \begin{cases} \mu_0 & \text{if } t \leq t_{\text{thre}} \\ 0 & \text{if } t > t_{\text{thre}} \end{cases} \tag{7}$$

where $\mu_0$ is the consistency weight. If $t > t_{\text{thre}}$, we only consider the diffusion optimization as existing works (Zhao et al., 2023a). if $t \leq t_{\text{thre}}$, we incorporate uncertainty-aware reward modeling.

**Discussion. 1). Why set timestep threshold $t_{\text{thre}}$ for $\mu$?** Considering that when $t$, *i.e.*, $t_1$ and $t_2$, is large, $z_t$ approaches the random noise $\epsilon$, leading to more diverse recovered $\hat{x}_0$ (see Fig. 1). In such cases, it becomes difficult to ensure that two outputs with identical conditions remain close, which is necessary for our uncertainty estimation, as it requires visually similar inputs to evaluate the reward model. In practise, we, therefore, empirically set a threshold to avoid the large timesteps for the reward modeling. We have conducted the experiment on the choice of $t_{\text{thre}}$ (see Section 4.3). **2). What is the advantage of uncertainty regularization in the reward feedback?** The advantage of uncertainty regularization in the reward feedback is that it helps to mitigate the adverse effects of imprecise feedback from the reward model. Specifically, by incorporating uncertainty estimation, the proposed method, Ctrl-U, can adaptively adjust the weights of the rewards during training. Rewards with lower uncertainty are given higher loss weights, ensuring that more reliable feedback

has a stronger influence on the training process. Conversely, rewards with higher uncertainty are given reduced weights, allowing for greater variability and preventing the model from being overly influenced by potentially inaccurate feedback. This adaptive rectification of rewards based on their uncertainty improves the consistency and reliability of the training process. As a result, the model becomes more robust to the diverse and sometimes unpredictable nature of newly generated data, leading to better controllability and generation quality.

## 4 EXPERIMENT

### 4.1 SETTINGS

**Datasets.** Our experiments are conducted using three datasets: ADE20K (Zhou et al., 2017; 2019), COCOStuff (Caesar et al., 2018) and MultiGen-20M dataset (Qin et al., 2023), adhering to the dataset construction principles of Controlnet++ (Li et al., 2024a). More specifically, we utilize the ADE20K and COCOStuff for the segmentation task. For Hed and Lineart conditions, we utilize pre-trained models, which is the same as ControlNet++, to generate annotations for the MultiGen-20M dataset. Considering that existing datasets usually include masks for pixels with unknown depth values, we adapt the MultiGen20M depth dataset in a manner similar to the dataset construction method used by ControlNet (Zhang et al., 2023a). For datasets without image captions, we use the captions generated by MiniGPT-4 (Zhu et al., 2023). The training and inference are conducted at a resolution of 512x512 for all datasets and methods.

**Evaluation and Metrics.** To evaluate semantic segmentation and depth map controls, we adopt mIoU and RMSE as the metrics, respectively. For the Hed edge and Lineart tasks, we calculate SSIM to compare the difference between the extracted control and the ground truth. For Sampling, we employ the UniPC (Zhao et al., 2023b) sampler, implementing 20 denoising steps to generate images using the original text prompts in accordance with ControlNet v1.1 (Zhang et al., 2023a), without incorporating any negative prompts. For comparative methods, we utilized their publicly available code to generate images, ensuring fairness by adhering to their original inference settings.

**Implementation Details.** In our experiments, we first fine-tune the pre-trained ControlNet model to convergence, using Adam as the optimizer with a learning rate of 1e-5, weight decay of 1e-2, and momentum of 0.9. Then, we use the same optimization settings to perform 10k iterations of uncertainty-aware reward fine-tuning. It is worth noting that we employ an one-step efficient reward strategy (Li et al., 2024a) to enhance training efficiency. Following the settings of previous work, we choose slightly weaker models as the reward model and stronger models for evaluation to guarantee fairness in assessment. See also the supplementary material for detailed reward model settings.

**Baselines.** In our evaluation, we mainly compare with competitive methods, including ControlNet++ (Li et al., 2024a), T2I-Adapter (Mou et al., 2024), ControlNet v1.1 (Zhang et al., 2023b), GLIGEN Li et al. (2023), Uni-ControlNet (Zhao et al., 2023a), and UniControl (Qin et al., 2023). All of the aforementioned methods have been fine-tuned on datasets across various tasks, and ControlNet++ introduces reward-based post-training on the foundation of ControlNet. These models perform well in controllable text-to-image generation, providing open-source model weights for re-implementation. For fair comparison, all models are tested under the same image conditions and text prompts. Although these methods utilize the user-friendly SD1.5 for controllable text-to-image generation, recent advancements have led to the adoption of SDXL (Podell et al., 2023) by several methods. Accordingly, we also present controllability results for ControlNet-SDXL and T2I-Adapter-SDXL. Since currently there are no officially released models, we deploy the third-party ControlNet-SDXL for the following experiments.

### 4.2 EXPERIMENTAL RESULTS

**Comparison of Controllability.** We present the results on five benchmarks in Table 1. Our uncertainty-aware method Ctrl-U significantly outperforms the previous state-of-the-art method ControlNet++ (Li et al., 2024a) by 6.53% in ADE20K and 8.65% in MultiGen20M depth, respectively. Regarding Hed and Lineart edge, the model with uncertainty regularization has obtained +3.76% and +1.06% increase on SSIM. The most significant improvement is observed in COCO-Stuff for segmentation masks, with an increase of 44.42%. This performance enhancement confirms that reducing the adverse effects of imprecise feedback from the reward model can significantly

Table 1: **Controllability comparison under various conditional controls and datasets.** ↑ denotes higher result is better, while ↓ indicates lower is better. '-' signifies the absence of a publicly available model for testing. The best result in each column is marked **bold** and the second is underlined. We generate four groups of png images and report their average result to reduce random errors.

| Condition (Metric) Dataset | T2I Model | Seg. Mask (mIoU ↑) | | Hed Edge (SSIM ↑) | LineArt Edge (SSIM ↑) | Depth Map (RMSE ↓) |
|---|---|---|---|---|---|---|
| | | ADE20K | COCO-Stuff | MultiGen-20M | MultiGen-20M | MultiGen-20M |
| ControlNet | SDXL | - | - | - | - | 40.00 |
| T2I-Adapter | SDXL | - | - | - | 0.6394 | 39.75 |
| T2I-Adapter | SD1.5 | 12.61 | - | - | - | 48.40 |
| Gligen | SD1.4 | 23.78 | - | 0.5634 | - | 38.83 |
| Uni-ControlNet | SD1.5 | 19.39 | - | 0.6910 | - | 40.65 |
| UniControl | SD1.5 | 25.44 | - | 0.7969 | - | 39.18 |
| ControlNet | SD1.5 | 32.55 | 27.46 | 0.7621 | 0.7054 | 35.90 |
| ControlNet++ | SD1.5 | 43.64 | 34.56 | 0.8097 | 0.8399 | 28.32 |
| Ctrl-U (Ours) | SD1.5 | **46.49** | **49.91** | **0.8401** | **0.8488** | **25.86** |

Table 2: **FID (↓) comparison under various conditional controls and datasets.** '-' signifies the absence of a publicly available model for testing. The best result in each column is marked **bold** and the second is underlined. We generate four groups of png images and report their average result to reduce random errors.

| Method | T2I Model | Seg. Mask | | Hed Edge | LineArt Edge | Depth Map |
|---|---|---|---|---|---|---|
| | | ADE20K | COCO-Stuff | MultiGen-20M | MultiGen-20M | MultiGen-20M |
| Gligen | SD1.4 | 33.02 | - | - | - | 18.36 |
| T2I-Adapter | SD1.5 | 39.15 | - | - | - | 22.52 |
| UniControlNet | SD1.5 | 39.70 | - | 17.08 | - | 20.27 |
| UniControl | SD1.5 | 46.34 | - | 15.99 | - | 18.66 |
| ControlNet | SD1.5 | 33.28 | 21.33 | 15.41 | 17.44 | 17.76 |
| ControlNet++ | SD1.5 | 29.49 | 19.29 | 15.01 | 13.88 | 16.66 |
| Ctrl-U (Ours) | SD1.5 | **28.61** | **15.79** | **11.59** | **11.99** | **15.48** |

improve the model controllability. We use the same hyper-parameter settings as those in the ControlNet++ experiments, verifying the effectiveness of our uncertainty-aware reward modeling.

**Comparison of Image Quality.** We employed the Fréchet Inception Distance (FID) (Heusel et al., 2017) to measure the distribution distance between generated and real images in Table 2. We could observe that, compared with existing methods, Ctrl-U exhibits superior FID values in various conditional generation tasks. Notably, in the COCO-Stuff for segmentation masks and MultiGen20M for Hed edge, Ctrl-U achieves impressive improvements of 18.14% and 22.74%, respectively. This significant boost indicates that our method not only enhances the controllability but also improves the quality of image generation, underscoring the efficacy of our proposed uncertainty-aware reward modeling in adaptively adjusting the weights of the rewards during training.

**Comparison of CLIP Score.** To assess the potential impact on text controllability, we reported the CLIP-Score metrics across various conditional generation tasks in Table 3. We discovered that Ctrl-U generally exhibits comparable or superior CLIP-Score results in most cases, suggesting that our approach not only enhances the controllability of conditional controls but also maintains the original proficiency of text-to-image generation. To ensure a more comprehensive comparison, we re-implemented the CLIP-Score using the official checkpoint for ControlNet++ (Li et al., 2024a), and marked the results with * and gray in Table 3.

**Qualitative Analysis.** We show the visual comparisons between our Ctrl-U and previous state-of-the-art methods across various conditional scenarios in Fig. 3. Given the same text prompts and image-based conditions, we observe that existing methods usually generate images with areas that do not align with the image conditions. Other methods, taking the segmentation mask generation task as an example, produce content on the building that is irrelevant to the provided condition, reflecting relatively poor controllability. Similarly, under edge and depth conditions, other methods fail to accurately represent the nuances of various image controls. In contrast, images generated by Ctrl-U exhibit better consistency with the input conditions.

**Human Evaluation.** Following ControlNet++ (Li et al., 2024a), we choose four types of conditional controls (segmentation masks, Hed edge, Lineart edge, and depth conditions) for human evaluation.

Table 3: CLIP-score ($\uparrow$) comparison under various conditional controls and datasets. '-' signifies the absence of a publicly available model for testing. The best result in each column is marked **bold** and the second is underlined. * and gray denote the CLIP-score we re-implemented using the checkpoints provided by Controlnet++. We generate four groups of png images and report their average result to reduce random errors.

| Method | T2I Model | Seg. Mask | | Hed Edge | LineArt Edge | Depth Map |
|---|---|---|---|---|---|---|
| | | ADE20K | COCO-Stuff | MultiGen-20M | MultiGen-20M | MultiGen-20M |
| Gligen | SD1.4 | 31.12 | - | - | - | 31.48 |
| T2I-Adapter | SD1.5 | 30.65 | - | - | - | 31.46 |
| UniControlNet | SD1.5 | 30.59 | - | 31.94 | - | 31.66 |
| UniControl | SD1.5 | 30.92 | - | 32.02 | - | 31.68 |
| ControlNet | SD1.5 | 30.16 | 31.18 | 31.46 | 31.26 | **32.11** |
| ControlNet++ | SD1.5 | **31.96** | 13.13 | **32.05** | **31.95** | 32.09 |
| ControlNet++* | SD1.5 | 31.24 | 30.93 | 30.49 | 30.34 | 30.02 |
| Ctrl-U (Ours) | SD1.5 | 31.26 | **31.23** | **32.05** | 31.87 | 31.72 |

Table 4: Results of Human Evaluation

| Evaluation Category | Ours | ControlNet++ | ControlNet | UniControl | UniControlNet |
|---|---|---|---|---|---|
| Image-Condition Alignment | **72.5%** | 7.5% | 6.2% | 7.5% | 12.5% |
| Image Quality | **56.2%** | 10.0% | 15.0% | 15.0% | 13.7% |
| Image-Text Alignment | **50.0%** | 27.5% | 22.5% | 7.5% | 12.5% |

We invite 20 participants to rank each result based on three distinct criteria. According to the average rankings shown in Table 4, users prefer the results produced by our method over those generated by the comparative methods.

### 4.3 Ablation Studies and Further Discussion

**Design of Uncertainty Estimation.** We present an ablation study on the design of the two-time generation in Table 5a. To mitigate the adverse effects of inaccurate rewards, we forward the identical input condition twice with different noise timestep to estimate uncertainty. Specifically, we adjust the disparity between the two timesteps, *i.e.*, $|t_1 - t_2|$. A short interval, such as $t_1 = t_2$, where the only randomness stems from resampled noise $\epsilon$, limits the diversity of the generated images. Considering the generated images are too similar, the reward discrepancy is small, and could not serve as uncertainty indicator. Conversely, a long interval indicates a significant gap between the two noisy latents, which, in turn, increase the generation discrepancy. Too large discrepancy also impacts the accurate uncertainty estimation, and thus compromises reward modeling. When $|t_1 - t_2| = 1$, the model achieves the optimal FID and relatively strong mIoU and CLIP-score.

**Impact of Different Timestep Threshold $t_{\text{thre}}$.** We investigate the impact of varying timestep threshold $t_{\text{thre}}$ (see Eq. 7) in Table 5b. We observe that, as the $t_{\text{thre}}$ increases, the scope of our uncertainty-aware reward rectification broadens, leading to improved mIoU. However, while increasing the rectification range can improve alignment between generated images and input conditions, setting the threshold too high will negatively affect image quality, resulting in higher FID scores. We find that the $t_{\text{thre}} = 400$ setting strikes an optimal balance, facilitating uncertainty estimation and regularization to adaptively correct the reward learning process.

**Impact of Regularization Weight $\lambda$.** We conducted a study to analyze the impact of varying uncertainty regularization weight $\lambda$ in Table 5c. When using a low value for $\lambda$, which imposes a minimal penalty for high uncertainty, it leads to excessively high uncertainty values across all samples. Conversely, a high value of $\lambda$ imposes a strong penalty on high uncertainty, suppressing the impact of uncertainty during reward fine-tuning. The setting with $\lambda = 1$ achieves an optimal balance. This configuration provides enough model capacity to accurately fit reward feedback while preventing overfitting to inaccurate rewards. As a result, the adaptive rectification of rewards based on uncertainty regularization improves the consistency and reliability of the training process.

**Impact of the Consistency Weight $\mu_0$.** We conduct an ablation study on the consistency weight $\mu_0$ as shown in Table 5d. We notice that the model shows insensitivity to the $\mu_0$ change in terms of CLIP-score. Setting $\mu_0$ to 1 yields the highest mIoU but at the cost of FID performance. This is due to the high weight of the consistency loss, which biases model training towards reward learning,

Table 5: Ablation studies on the ADE20K dataset. We report mIoU, FID and CLIP-score to evaluate controllability and image quality respectively. (a) We show the impact of the interval between $t_1$ and $t_2$. We find that a short interval limits the diversity of the generated images, making it challenging to estimate uncertainty. In contrast, a long interval impacts accurate uncertainty estimation and thus compromises reward modeling. (b) We study the timestep threshold $t_{\text{thre}}$. We find that the $t_{\text{thre}} = 400$ setting strikes an optimal balance, facilitating uncertainty estimation and regularization to adaptively correct the reward learning process. (c) The impact of the regularization weight $\lambda$. With $\lambda = 1$, we obtain best results. This setting provides enough model capacity to accurately fit reward feedback while preventing overfitting to inaccurate rewards. (d) The impact of the consistency weight $\mu_0$. Considering the balance between diffusion training and uncertainty-aware reward learning, we set $\mu_0 = 0.1$ to ensure enhanced controllability while improving generation quality.

<table>
<tr><td colspan="7" align="center">(a)</td><td colspan="4" align="center">(c)</td></tr>
<tr><td>$|t_1 - t_2|$</td><td>0</td><td>1</td><td>3</td><td>5</td><td>7</td><td>9</td><td>$\lambda$</td><td>0.05</td><td>0.1</td><td>1</td></tr>
<tr><td>mIoU</td><td>45.33</td><td>46.49</td><td>**46.72**</td><td>45.94</td><td>45.11</td><td>44.88</td><td>mIoU</td><td>43.06</td><td>43.47</td><td>**46.49**</td></tr>
<tr><td>FID</td><td>29.01</td><td>**28.61**</td><td>29.01</td><td>29.81</td><td>29.10</td><td>28.94</td><td>FID</td><td>30.15</td><td>30.60</td><td>**28.61**</td></tr>
<tr><td>CLIP-score</td><td>30.92</td><td>**31.26**</td><td>31.01</td><td>31.06</td><td>30.93</td><td>31.09</td><td>CLIP-score</td><td>30.88</td><td>30.80</td><td>**31.26**</td></tr>
<tr><td colspan="7" align="center">(b)</td><td colspan="4" align="center">(d)</td></tr>
<tr><td>$t_{\text{thre}}$</td><td>200</td><td>300</td><td>400</td><td>500</td><td>600</td><td>700</td><td>$\mu_0$</td><td>0.05</td><td>0.1</td><td>1</td></tr>
<tr><td>mIoU</td><td>43.82</td><td>43.89</td><td>46.49</td><td>46.72</td><td>47.92</td><td>**50.11**</td><td>mIoU</td><td>42.37</td><td>46.49</td><td>**49.48**</td></tr>
<tr><td>FID</td><td>29.70</td><td>30.34</td><td>**28.61**</td><td>31.04</td><td>32.98</td><td>34.21</td><td>FID</td><td>30.16</td><td>**28.61**</td><td>30.89</td></tr>
<tr><td>CLIP-score</td><td>31.13</td><td>30.68</td><td>**31.26**</td><td>31.07</td><td>31.01</td><td>30.98</td><td>CLIP-score</td><td>31.02</td><td>**31.26**</td><td>30.84</td></tr>
</table>

thereby enhancing controllability but compromising generation quality. When $\mu_0 = 0.1$, FID performance reaches its optimum, and mIoU remains strong, effectively balancing diffusion training and uncertainty-aware reward learning without causing the uncertainty to either vanish or explode.

**Efficacy of Uncertainty-aware Reward Modeling.** We validate the superiority of our learnable uncertainty-aware reward method by comparing it with the vanilla reward learning. As depicted in Fig. 4, under the identical input conditions and timesteps, although the generated images align with the given conditions, reward learning without uncertainty produces inaccurate predictions. On the contrary, our uncertainty-aware reward fine-tuning generates higher-quality images. Additionally, the reward model can accurately extract conditions from these images. This improvement is attributed to the reward uncertainty, which adaptively rectifies inaccurate rewards and improves the reliability of the training process. Specifically, rewards with lower uncertainty are given higher loss weights, enhancing the influence of precise feedback in the training process. Conversely, rewards with higher uncertainty are assigned lower weights, leading to greater variability and ensuring the model is not excessively influenced by potentially inaccurate feedback. Consequently, the model becomes more robust to the diversity of the newly generated images, enhancing both controllability and the quality of generation.

## 5 CONCLUSION

In this work, we focus on the challenge of ensuring both high fidelity and semantic alignment in conditional image generation. We highlight a significant limitation in existing methods that use pre-trained reward models for enforcing alignment. Yet these models often provide inaccurate feedback when encountering diverse, newly generated data, which can negatively impact the training process. To alleviate this issue, we propose an uncertainty-aware reward modeling approach, termed Ctrl-U, which incorporates uncertainty estimation and adaptive regularization. This method assigns higher loss weights to rewards with lower uncertainty and reduces the weights for highly uncertain rewards, thereby enhancing the consistency and reliability of the training process. Our extensive experiments across five benchmarks on three datasets, *i.e.*, ADE20k, COCO-Stuff, and MultiGen-20M, validate the effectiveness of our methodology in improving controllability and generation quality. Additionally, Ctrl-U shows robust scalability, performing well across various conditional scenarios, including segmentation masks, edges, and depth conditions. These results indicate the potential of our method to contribute to advancements in conditional image generation.

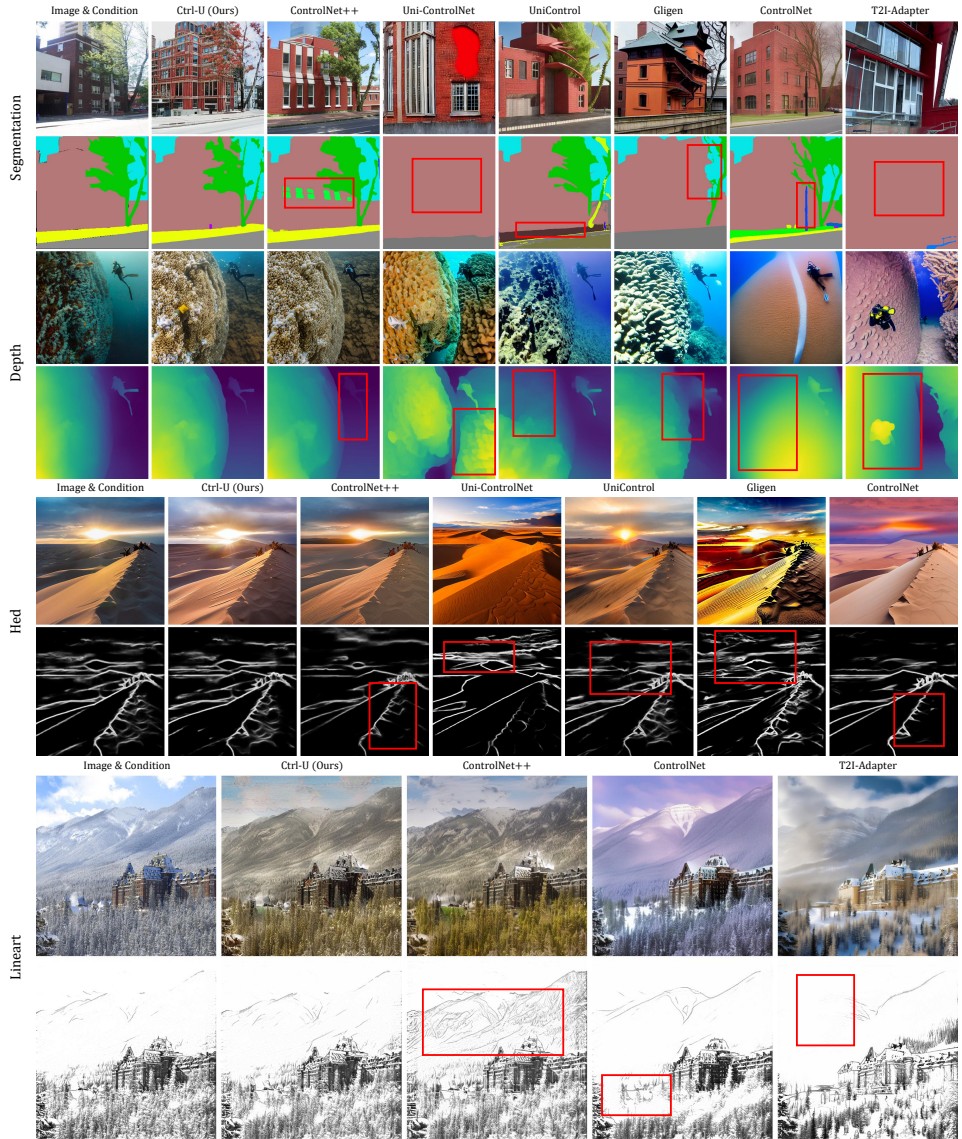

Figure 3: Qualitative comparisons with different conditional controls on unseen test images. We observe that our generated image preserves condition alignment with good visual quality. Some models do not have open-source weights for Hed or Lineart condition, and thus we skip them.

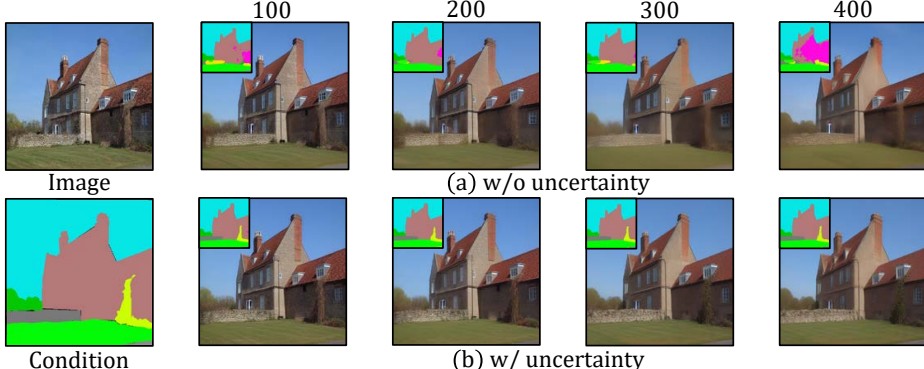

Figure 4: Ablation study on the uncertainty-aware reward modeling. Here, we show the recovered test image with different denoising timesteps. Since we mitigate the negative impact of noisy rewards, our output maintains consistent semantic conditions with fewer blurred areas.

ACKNOWLEDGEMENT

This work is supported by the University of Macau Start-up Research Grant SRG2024-00002-FST and Multi-Year Research Grant MYRG-GRG2024-00077-FST-UMDF.

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

## 5.1 MORE IMPLEMENTATION DETAILS.

**Dataset Details.** We use the ADE20K dataset for segmentation masks, which includes 20,210 images in the training set and 2,000 images in the validation set. The dataset contains 150 types of segmentation annotations, covering most objects found in daily life scenes. Since ADE20K (Zhou et al., 2017; 2019) lacks text prompts, we follow ControlNet++ (Li et al., 2024a) and use MiniGPT-4 (Zhu et al., 2023) to generate image captions by instructing: "Please briefly describe this image in one sentence". Similarly, COCO-Stuff provides segmentation annotations, with 118,287 images in the training set and 5,000 in the validation set. It is challenging to find datasets with accurate annotations for Hed and LineArt edge. To tackle this issue, we adhere to the dataset construction principles of ControlNet (Zhang et al., 2023a) and train our model using the MultiGen20M (Qin et al., 2023) dataset, which includes annotations generated by the model. The images and their corresponding captions are displayed in Fig. 5.

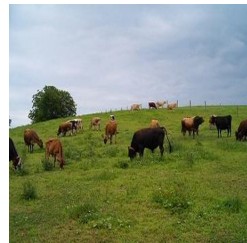 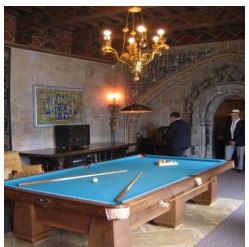 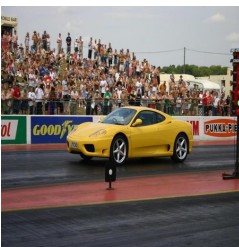 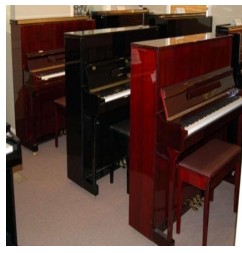

A group of cows grazing on a lush green hillside.

A wooden billiards table with a green felt surface and ornate wooden trim, surrounded by a stone wall with a chandelier hanging from it.

A yellow sports car is driving on a track in front of a large crowd of people watching.

A group of upright pianos arranged in a row in a room.

Figure 5: Text prompts generated by MiniGPT-4

**Reward Model Details.** We provide details of the models and evaluation metrics in Table 6. Considering that the Hed and Lineart edge extraction models are neural networks without non-differentiable operations, we achieve differentiability by modifying the forward code. We have improved the reward fine-tuning process by the proposed uncertainty-aware reward modeling, resulting in significant enhancement in controllability and generation quality across four control conditions. Our research will expand to encompass additional control scenarios, such as human pose and scribbles.

Table 6: Details of the reward model, evaluation model, and training loss under different conditional controls. ControlNet* indicates that we utilize the same model for condition extraction as employed in ControlNet (Zhang et al., 2023a).

|  | Seg. Mask | Depth Edge | Hed Edge | LineArt Edge |
|---|---|---|---|---|
| **Reward Model (RM)** | UperNet-R50 | DPT-Hybrid | ControlNet* | ControlNet* |
| **RM Performance** | ADE20K (mIoU): 42.05 | NYU (AbsRel): 8.69 | - | - |
| **Evaluation Model (EM)** | Mask2Former | DPT-Large | ControlNet* | ControlNet* |
| **EM Performance** | ADE20K (mIoU): 56.01 | NYU (AbsRel): 8.32 | - | - |
| **Consistency Loss** | CrossEntropy Loss | MSE Loss | MSE Loss | MSE Loss |

## 5.2 EXPLANATION OF UNCERTAINTY VISUALIZATION

We present additional qualitative results in Fig. 6. As shown in Fig. 6b, we could observe that the high variance usually exists in the area with incorrect reward. The proposed rectified loss assigns different thresholds to different areas. For example, for the location with coherent rewards between two generations, the variance regularization drives the model trust reward feedback. For the area with ambiguous rewards, the variance regularization drives the model to neglect reward feedback. Our uncertainty-aware framework adaptively adjusts the loss weights of different reward feedback, facilitating more effective reward optimization.

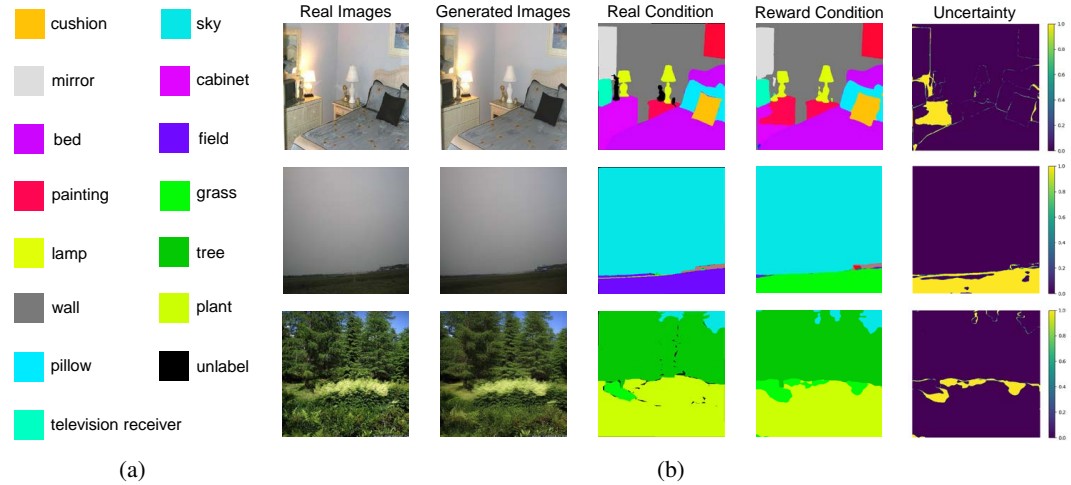

(a)

(b)

Figure 6: Illustration of the estimated uncertainty of inaccurate reward. **(a)** Detailed category illustration. **(b)** The areas in the reward condition, where generated images align with the real condition, but receive incorrect rewards, obtain large value of the prediction variance. Meanwhile, we could observe that the high-variance area has considerable overlaps with the wrong segmentation feedback from the reward model.

## 5.3 MORE VISUALIZATION

More visualization results across different conditional controls for our image generation on the validation set are shown in Figures 7, 8, 9, 10. We observe that our uncertainty-aware framework is capable of generating diverse images that align well with the given inputs across semantic masks, edge, and depth conditions. Attributed to our proposed uncertainty estimation and regularization, the adaptive rectification of inaccurate rewards enhances the diversity of the generated data and its alignment with the given conditions.

Image & Condition                          Generated Images & Extracted Conditions

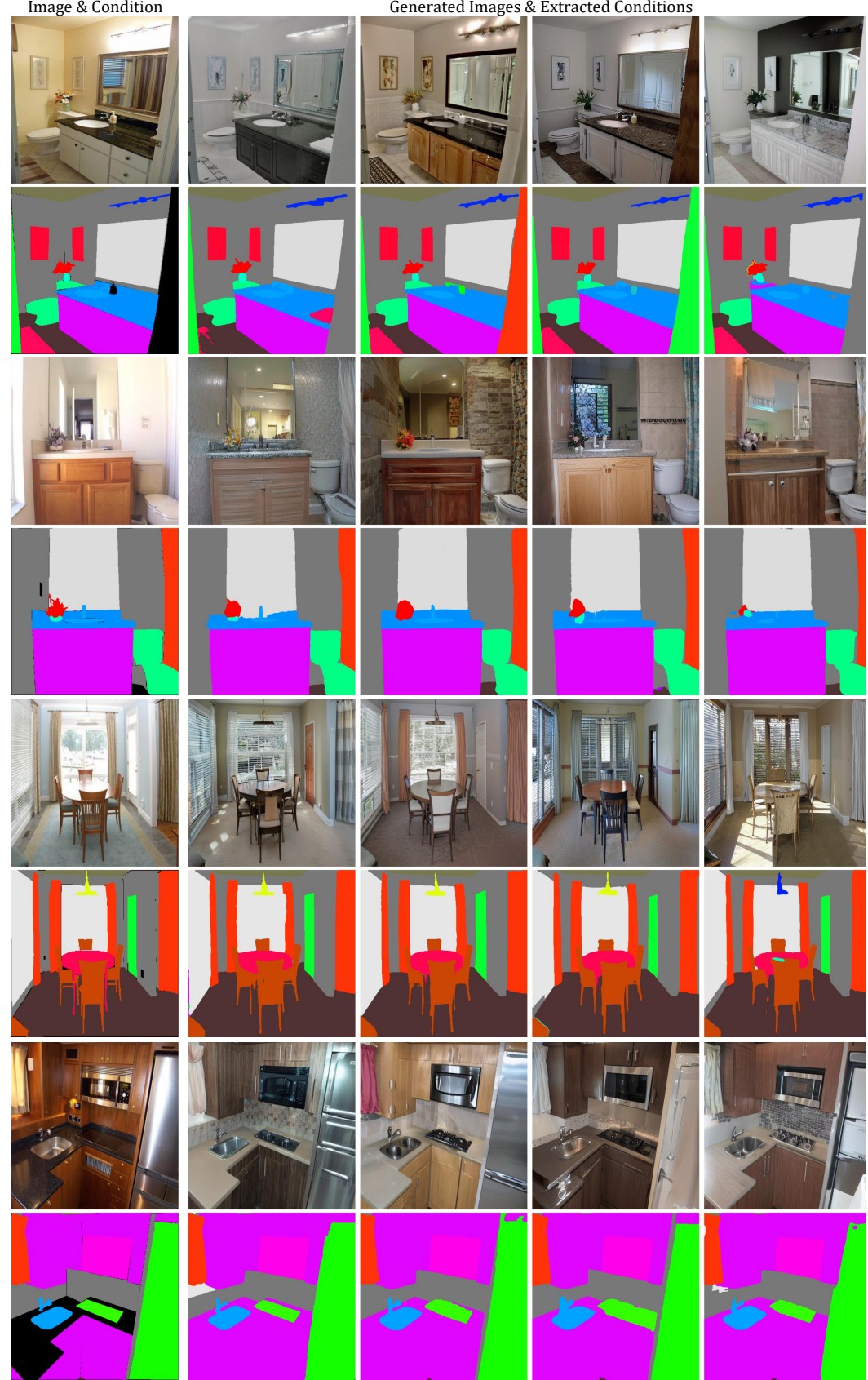

Figure 7: More visualization results of ours on unseen test images (Segmentation Mask)

Image & Condition          Generated Images & Extracted Conditions

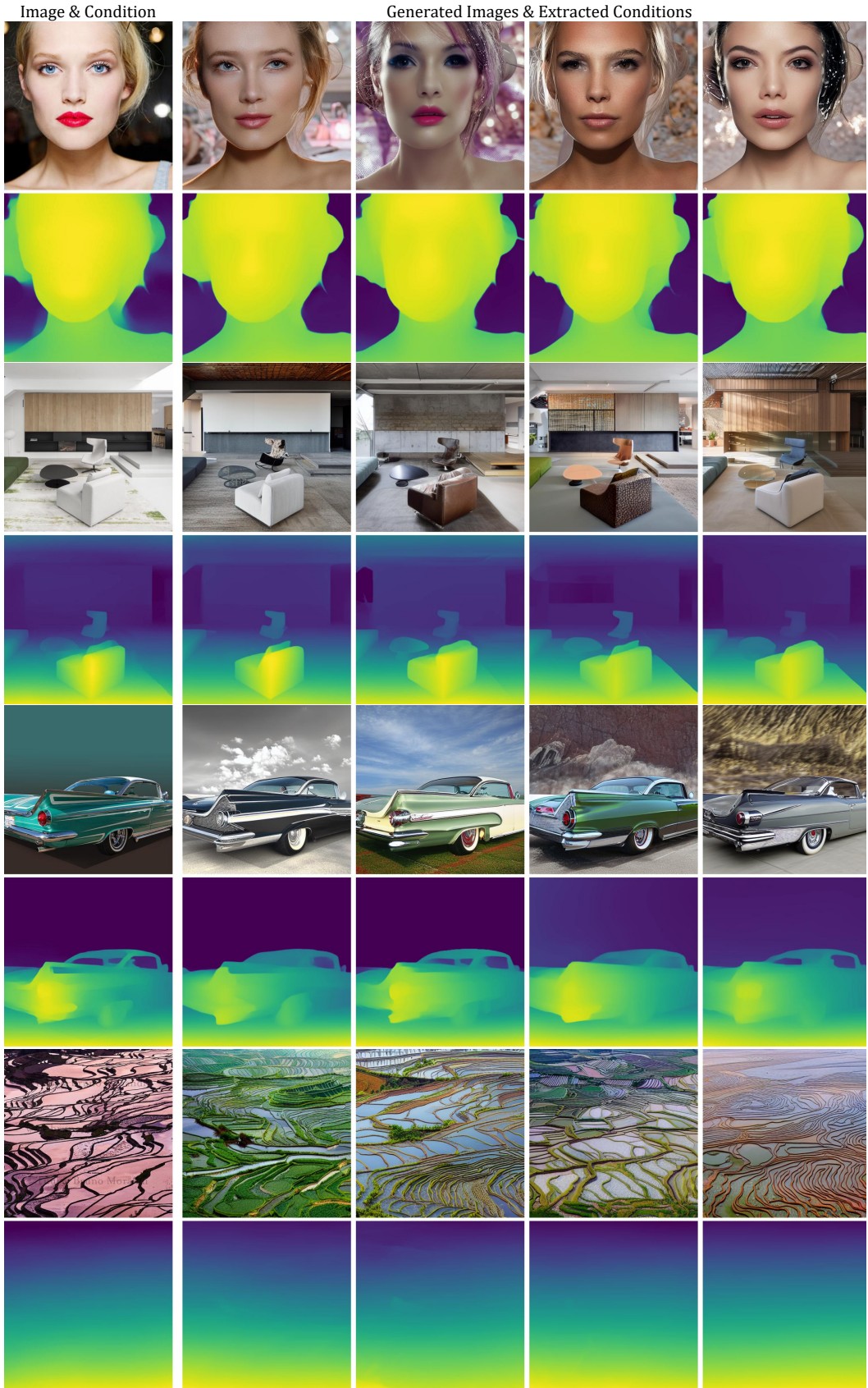

Figure 8: More visualization results of ours on unseen test images (Depth Map)

Image & Condition                    Generated Images & Extracted Conditions

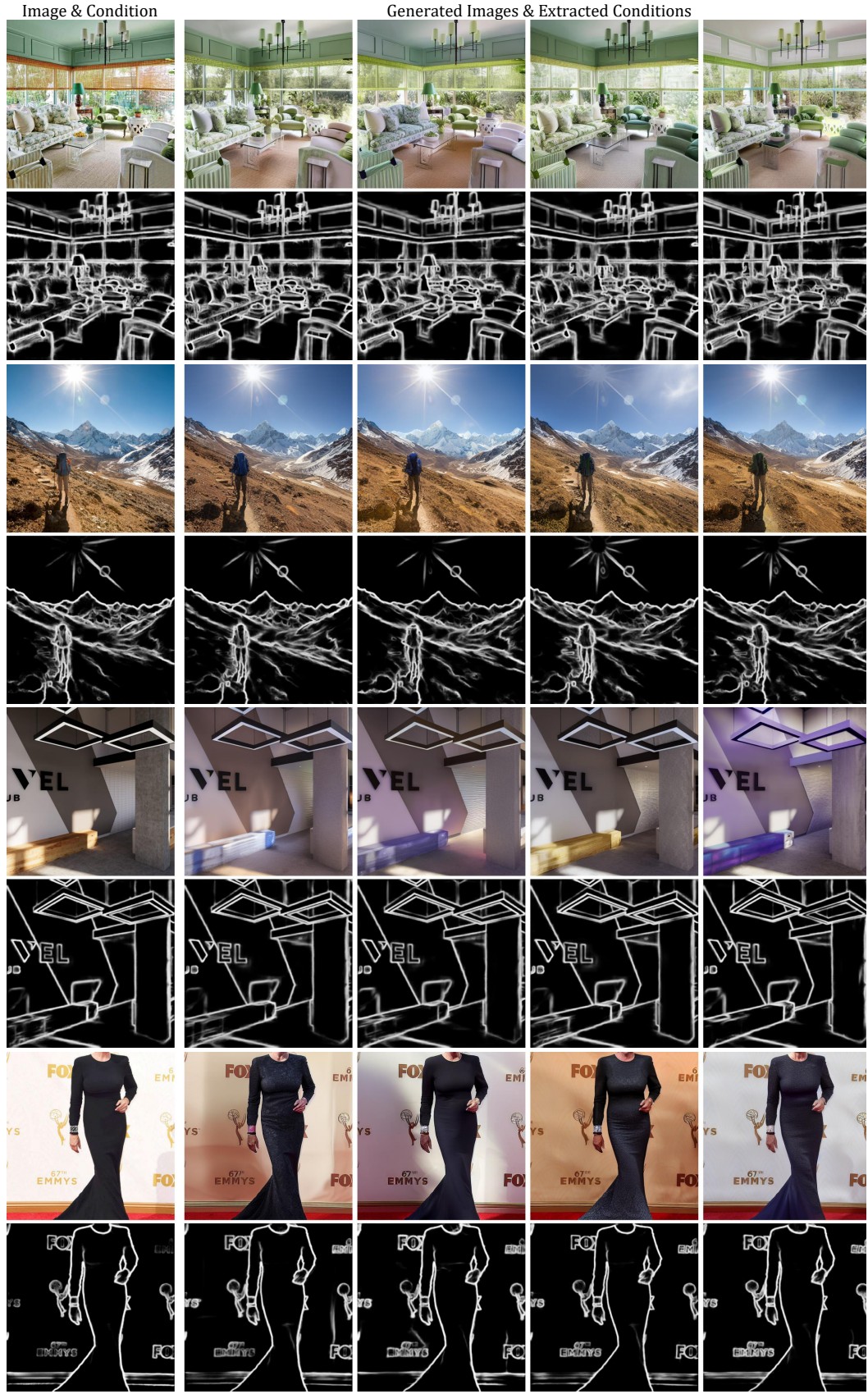

Figure 9: More visualization results of ours on unseen test images (Hed Edge)

Image & Condition               Generated Images & Extracted Conditions

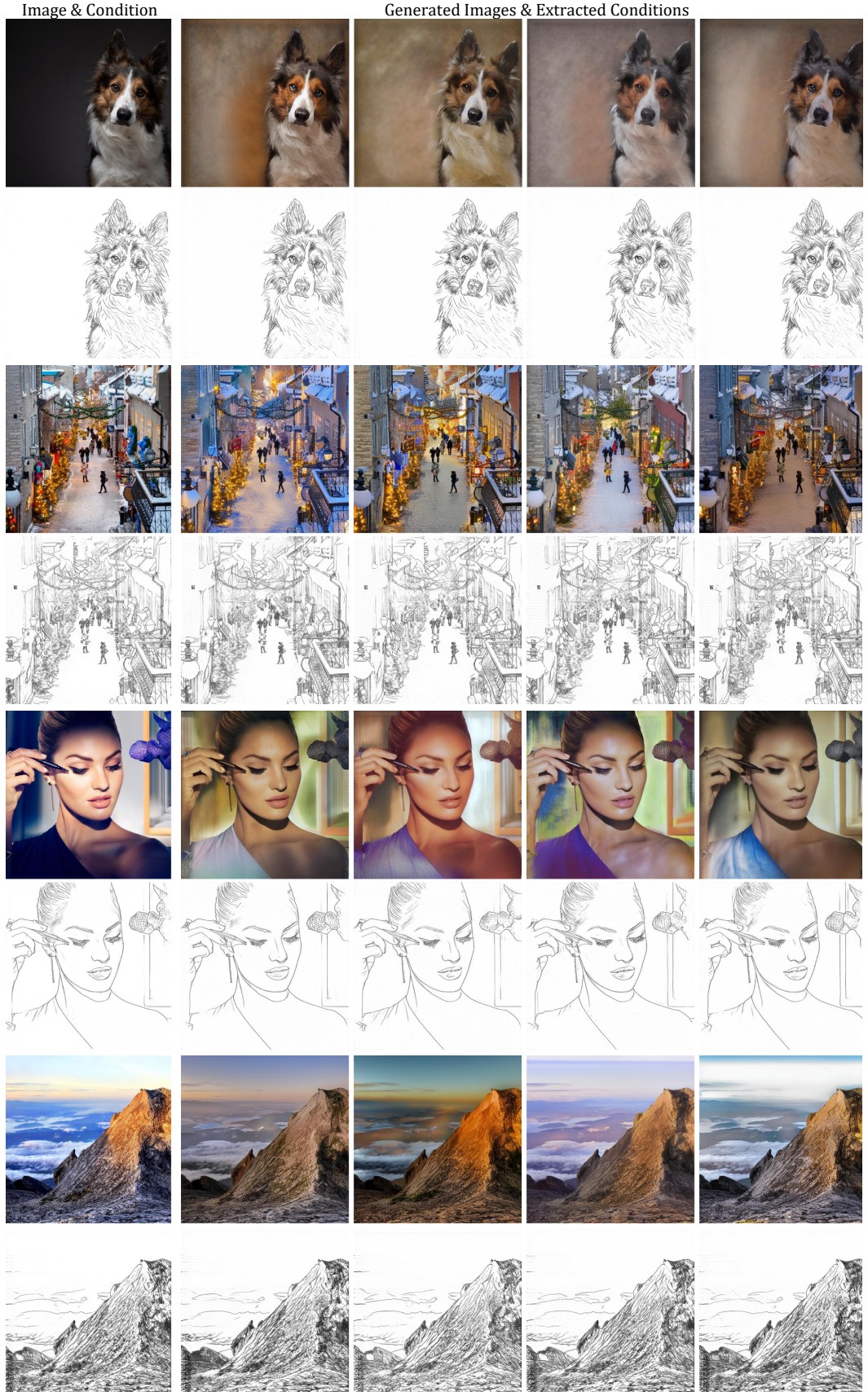

Figure 10: More visualization results of ours on unseen test images (LineArt Edge)

