# OpenReview forum: "Ctrl-U: Robust Conditional Image Generation via Uncertainty-aware Reward Modeling"
_ICLR.cc/2025/Conference — ICLR 2025 Poster_

### Official Review · Reviewer_xjaQ · 2024-11-01

**Soundness:** 3
**Presentation:** 3
**Contribution:** 2
**Rating:** 6
**Confidence:** 3

**Summary:**

This paper made improvements to the controllability of diffusion-based conditional generative models by introducing the concept of uncertainty.
The authors designed loss functions for probabilistic and non-probabilistic conditions by assessing the differences between images in two inference processes.
The authors claim that they have achieved SOTA performance on both objective and subjective metrics.

**Strengths:**

I appreciate the motivation behind this paper, and I believe the method presented is quite sound.

**Weaknesses:**

1. We found that the loss function proposed in this paper can lead to artifacts, such as the linerart in Fig 3. The proposed method can achieve alignment between the outputs and the conditions, but the authors do not discuss how to maintain the visual plausibility of the generated results while ensuring this alignment.

2. In Tab.5a, there seems to be no clear evidence of what the optimal value of delta t is. We hope the authors can further discuss the choice of delta t. I am curious about why a relatively small delta t (1% of total 1000 timesteps) can encourage the diversity of generated samples.

3. From Fig.2, it can be seen that the gradient obtained from 2 different timesteps is back-propagated, and the parameters of the UNet will be also optimized. We hope the authors can provide some discussion on the training computational cost. Have the authors tried to only optimize the parameters of the control net?

4. This point does not affect our rating, but we want to know why the ControlNet method on SDXL in Tab.1 performs significantly worse than on SD1.5.

**Questions:**

Please refer to Weaknesses.

---

> ### Author Response · Authors · 2024-11-22
> **Response for Reviewer xjaQ**
>
> Dear Reviewer xjaQ
>
> Thank you for your thorough review of our manuscript and for recognizing our efforts. Your insights have been instrumental in enhancing the quality of our work, and we are pleased to discuss several research points with you.
>
> **Weaknesses:**
>
> (1) The visual quality of generated images by Ctrl-U in Figure 3.
>
> The issue you mentioned does indeed exist. It is worth noting that the Hed and LineArt  conditions in the MultiGen20M dataset are generated by the model (please refer to Appendix A.1 for details).
> When the local quality of the input condition is low, e.g., the given lineart condition in Fig. 3 fails to depict the clouds in the sky, maintaining alignment between the condition and the generated result can adversely affect the image quality.
> There is a balance problem between the fidelity of the generated images and their semantic alignment with the provided conditions. Although some visual implausibility remains, extensive experiments confirm the effectiveness of our method in balancing controllability and visual quality. As shown in Tables 1 and 2, our approach not only enhances controllability but also improves overall image quality across diverse conditional scenarios, achieving a lower Fréchet Inception Distance (FID) compared to existing methods.
>
> (2) Discussion of the choice of $\Delta t$ ($|t_1 - t_2|$).
>
> We apologize for the lack of clarity. As shown in Table 5 (a), the optimal value for $\Delta t$ ranges from 1 to 3. We find that when $\Delta t$ is large, the differences in the generated images become great, leading the calculated variance less meaningful. When $\Delta t$ is smaller and could induce slight jitter, the generated images are more similar, and the estimated uncertainty more accurately reflects the inaccurate reward.
>
> To further verify the optimal $\Delta t$, we conducted ablation studies across different datasets and tasks, and find that the optimal $\Delta t$ is not sensitive to different dataset and tasks.
>
>
> | Method             | Segmentation (ADE20K) | | | Segmentation (COCO-Stuff) | |  | Depth (MultiGen-20M) | | |
> |:---:|:---:|:---:|:---:| :---:|:---:|:---:| :---:|:---:|:---:|
> |                     | mIoU | FID    | CLIP  | mIoU  |  FID   |   CLIP | RMSE   | FID   | CLIP |
> |ControNet++| 43.64 | 29.49 | 31.24 | 34.56 | 19.29 |  30.93 |  28.32 |16.66 |**32.02** |
> | \|t1 − t2\| = 0 |45.33 |29.01  |30.92  | 45.36 |15.94 | 31.19| 26.93 |16.26 |31.67|
> | \|t1 − t2\| = 1 |46.49 | **28.61** | **31.26** | **49.91** |**15.79** | 31.23| 25.86 |**15.48** |31.72|
> | \|t1 − t2\| = 3| **46.72**| 29.01 |31.11  |48.01  |15.99 | 31.11 |**24.63**  |15.53 |31.17|
> |\|t1 − t2\| = 5 | 45.94| 29.21 |31.16 |47.74 |15.95 |**31.26**   |26.59  |15.93  |31.64|
> |\|t1 − t2\| = 9 | 44.88| 28.94 |31.19 |47.67 |15.96 | 31.19  |25.40  |15.79  |31.58|
>
> (3) Discussion of the training computational cost.
>
> Apologies for the Fig. 2. In fact, we froze the parameters of UNet and only optimized the parameters of ControlNet. We have updated Fig. 2 in the revised version. As for the training computational cost, on the H800 GPU, training for 10k steps with a batch size of 6 and a learning rate of 1e-5, the uncertainty-aware reward method requires 3.6 hours and 68.9GB of memory.
>
> (4) Table 1: Why does the ControlNet method perform significantly worse on SDXL than on SD1.5?
>
> Good question! Although SDXL [1] outperforms SD1.5 in text-to-image synthesis, ControlNet-SDXL fails to surpass ControlNet-SD1.5 in the realm of image-based controllable generation. Presently, unlike ControlNet-SD1.5, there is no officially released model of ControlNet-SDXL. Therefore, our evaluation of ControlNet-SDXL was conducted using a model obtained from the open-source community. The unavailability of training data in ControlNet [2] constrains the fine-tuning of downstream tasks for ControlNet-SDXL, thereby limiting its model performance.
>
> **References:**
>
> [1] Podell, D., English, Z., Lacey, K., Blattmann, A., Dockhorn, T., Müller, J., Penna, J., Rombach, R.: Sdxl: Improving latent diffusion models for high-resolution image synthesis. International Conference on Learning Representations (ICLR), 2024.
>
> [2] Zhang, L., Rao, A., Agrawala, M.: Adding conditional control to text-to-image diffusion models. International Conference on Computer Vision (ICCV), 2023.

---

> ### Author Response · Authors · 2024-11-22
> **Sincere Request for Further Discussions**
>
> Dear Reviewer xjaQ,
>
> Thanks again for your great efforts and constructive advice in reviewing this paper! With the discussion period drawing to a close, we expect your feedback and thoughts on our reply. We put a significant effort into our response, with several new experiments and discussions. We sincerely hope you can consider our reply in your assessment. We look forward to hearing from you, and we can further address unclear explanations and remaining concerns if any.
>
> Regards,
>
> Authors

---

### Official Review · Reviewer_4HGt · 2024-11-02

**Soundness:** 2
**Presentation:** 3
**Contribution:** 2
**Rating:** 6
**Confidence:** 3

**Summary:**

Targeting  conditional image generation, the authors proposed an uncertainty-aware reward modeling method, named Ctrl-U. This method includes uncertainty estimation and uncertainty-aware regularization, aimed at mitigating the negative effects of inaccurate feedback from the reward model. The authors use the prediction variance as an uncertainty indicator, and based on this, they adaptively adjust the reward during model training. Rewards with lower uncertainty are given higher loss weights, while those with higher uncertainty are given reduced weights, allowing for more variability.

**Strengths:**

1. This paper targets the uncertainty problem existing in training with reward models, which is critical and meaningful.

2. Despite increasing training time for each training step, the proposed method avoids incorporating additional modules to assess reward uncertainty.

**Weaknesses:**

1. Lack of novelty. The major contribution of this work is the methodology that generates two images, calculates their uncertainty, and reweights the reward loss. Such a solution seems to be trivial, double the time of each training step, and bring neglective quantitative improvement.
Maybe the author can report the computational cost required before and after using the proposed method.
In addition, why do not the authors collect a dataset composed of paired generated images and their uncertainty scores and then train a simple predictor to predict uncertainty for each image to remove the necessity to generate two images in one step? In my opinion, such an uncertainty predictor can also be adaptive to different generative models and can significantly improve the training efficiency compared to the proposed method. Since the predictor can be more lightweight than generative models and can be trained on image latents.

2. Unfair comparison. The method proposed in this paper is mainly used for post-training. In other words, authors need to pretrain a conditional model such as ControlNet before employing the proposed training losses (Line 292-295). However, during quantitative comparison, authors compare their methods with pretrained models. Such a comparison seems to be inappropriate. It may be better to report the performance improvement when applying the proposed method to different base models, showing the generalization ability of the proposed method.

**Questions:**

1. As specified in L215-217, authors stated that directly regressing uncertainty can incur indistinguishable predictions over samples. However, In L239-240, the proposed methods also require additional terms to prevent indistinguishable predictions. Considering that the proposed method can introduce double training time, what’s the major advantage of the proposed method?
2. I'm confused by the value of t in equ. 7. Specifically, how does t in equ.7 relate to t1 and t2 introduced in equ.5?

---

> ### Author Response · Authors · 2024-11-22
> **Response for Reviewer 4HGt (Part #1)**
>
> Dear Reviewer 4HGt
>
> We sincerely appreciate your constructive reviews. We hope our response can address your concerns.
>
> **Weaknesses:**
>
> (1-1) The contribution and computational cost of our work.
>
> 1. Our technical contributions.
>
> Our Ctrl-U is not just a trivial design, but brings new insight to the field of conditional generation. To the best of our knowledge, we are the first to observe the adverse effects of inaccurate reward feedback on conditional image generation, and introduce an uncertainty-aware reward modeling approach that can adaptively regularize the reward learning process. Extensive experiments validate that the quantitative improvements are significant. As shown in Table 1, Ctrl-U significantly outperforms the previous state-of-the-art method ControlNet++ [1] relatively by 6.53\%  on ADE20K and 8.65\%  on MultiGen20M depth, respectively. Regarding Hed and Lineart edge, the model with uncertainty regularization has obtained +3.76\%  and +1.06\%  increase on SSIM. The most significant improvement is observed in COCOStuff for segmentation masks, with an increase of 44.42\%.
>
> 2. The computational cost of our method.
>
> Thanks for your advice! It is very necessary to report the computational cost before and after using the proposed method. On the H800 GPU, training for 10k steps with a batch size of 6 and a learning rate of 1e-5, the uncertainty-aware reward method increases the average step time from 0.78s to 1.32s and memory usage from 38.9GB to 68.9GB.
> However, we argue that this increment in time and memory usage is justified because it significantly enhances the performance upper bound of training conditional generation models, as mentioned above. Simply finetuning for a longer time cannot achieve this level of improvement.
> Furthermore, there is no additional overhead during inference. We only need the trained encoders and diffusion model during testing, and the reward model is dropped.
>
> (1-2). Why not collect dataset and train an uncertainty predictor?
>
> 1. Collect dataset.
>
> Annotating uncertainty manually remains challenging, especially for the uncertainty score.
> For instance, some samples are ambiguous to machine but not for human, while some samples are easy for machine but challenging to human annotators. Therefore,
> current uncertainty learning [2-3] primarily employs Bayesian approaches to model uncertainty, i.e., the variance between two predictions.
>
> 2. Uncertainty quality.
>
> We have included uncertainty visualization in Appendix A.2, which shows that the estimated uncertainty by our method is highly correlated with incorrect rewards.
>
> 3. Other methods.
>
> Furthermore, the reviewer gccp suggested comparing REVAR [4], which uses auxiliary networks to predict uncertainty. Experiments validate that our method is more suitable for  uncertainty measurement on conditional generation. Please refer to our response to reviewer gccp's Q1 for further details.

---

> ### Author Response · Authors · 2024-11-22
> **Response for Reviewer 4HGt (Part #2)**
>
> (2) Unfair comparisons.
>
> 1. The detail of comparisons.
>
> We would like to clarify that all comparison methods in Tables 1-3 are finetuned across different tasks using the official pretrained models.
> We do not compare the pretrained models without fine-tuning.
> ControlNet++ [1] is a version of ControlNet with reward model for post-training, so we directly compare our approach with ControlNet++ rather than ControlNet.
>
> 2. The generalization of the proposed method to different base models.
>
> Thank you for suggesting applying the proposed method to different base models. We conduct an ablation study on the proposed method to different base models, e.g., SD1.5 and SD2.1, in the table below.
>
>
> | Method             |SD1.5| | | SD2.1 | |  |
> |:---:|:---:|:---:|:---:| :---:|:---:|:---:|
> |                     | mIoU | FID    | CLIP  | mIoU  |  FID   |   CLIP |
> |ControlNet| 32.55 |33.28 |30.16 |40.36 |30.95 |**30.88**|
> |ControlNet++ |44.64 |29.49 |31.24 |45.39 |29.19| 30.85|
> |Ctrl-U |**46.49** |**28.61** |**31.26** |**48.71** |**28.47** |30.76|
>
> As shown above, our uncertainty-aware reward method consistently attains better results than the existing models. This indicates the potential of our approach to achieve better performance on different base models.

---

> ### Author Response · Authors · 2024-11-22
> **Response for Reviewer 4HGt (Part #3)**
>
> **Questions:**
>
> (1) "As specified in L215-217, authors stated that directly regressing uncertainty ....."
>
> 1. In lines 215-217, how about regress uncertainty?
>
> We find that regressing uncertatinty without constriants usually leads to over-fitting.
> We also conduct a experiment to verify this point.
> As suggested by Reviwer gccp, we compare a regressing method REVAR, which requires an extra network to regress uncertainty.
> We observe the proposed method surpasses REVAR.
>
> 2. The main advantage of the proposed method.
>
> Our uncertainty mechanism is based on Bayesian network, which harnesses different predictions to capture the output variance. Similarly, the proposed method captures the uncertainty via variance between two predictions.
> Comparing to regression-based uncertainty without sufficient constraints, we models uncertainty by calculating the variance across predictions, which are with physical meaning.
> Although the proposed method increases computational overhead, the prediction variance effectively captures the inaccurate feedbacks, facilitating the optimization (in Tables 1-3).
> Please see the uncertainty visualization in Appendix A.2.
>
> 3. In L239-240, more explanation of $\lambda \cdot U$ in Eq. 4.
>
> The second term is to avoid the large value of uncertainty. We follow the existing design in [5].
> If we only consider the first term, the optimizer tends to always predict the larger uncertainty to minimize the loss. If we add the second term, it mitigate such large-uncertainty cases.
>
> (2) More explanation of $t$ in Eq. 5 and Eq. 7.
>
> Sorry for the confusion. In our context, $t$ denotes the general timestep, and therefore, it could represent both $t_1$ and $t_2$.
> Eq. 7 ensures that $t_1$ and $t_2$ are less than $t_{\text{thre}}$.
> We have revised the manuscript to provide a clearer explanation.
>
> **References:**
>
> [1] Ming Li, Taojiannan Yang, Huafeng Kuang, Jie Wu, Zhaoning Wang, Xuefeng Xiao, and Chen Chen. Controlnet++: Improving conditional controls with efficient consistency feedback. European Conference on Computer Vision (ECCV), 2024.
>
> [2] James Harrison, John Willes, and Jasper Snoek. Variational bayesian last layers. International Conference on Learning Representations (ICLR), 2024.
>
> [3] Junbo Li, Zichen Miao, Qiang Qiu, and Ruqi Zhang. Training bayesian neural networks with sparse subspace variational inference. International Conference on Learning Representations (ICLR), 2024.
>
> [4] Nishant Jain, Karthikeyan Shanmugam, and Pradeep Shenoy. Learning model uncertainty as variance-minimizing instance weights. International Conference on Learning Representations (ICLR), 2024.
>
> [5] Alex Kendall and Yarin Gal. What uncertainties do we need in bayesian deep learning for computer vision? Advances in neural information processing systems, 30, 2017.

---

> ### Author Response · Authors · 2024-11-22
> **Sincere Request for Further Discussions**
>
> Dear Reviewer 4HGt,
> Thanks again for your great efforts and constructive advice in reviewing this paper! With the discussion period drawing to a close, we expect your feedback and thoughts on our reply. We put a significant effort into our response, with several new experiments and discussions. We sincerely hope you can consider our reply in your assessment. We look forward to hearing from you, and we can further address unclear explanations and remaining concerns if any.
> Regards,
> Authors

---

> > ### Comment · Reviewer_4HGt · 2024-11-26
> >
> > Thank you for your detailed response, which addressed my concerns well.
> > I will raise my score accordingly.

---

### Official Review · Reviewer_gccp · 2024-11-06

**Soundness:** 3
**Presentation:** 3
**Contribution:** 3
**Rating:** 6
**Confidence:** 2

**Summary:**

This paper addresses the issue of conditional image generation. The authors identify a significant flaw in the perceptual loss used for conditional image generation, where it may provide incorrect supervisory signals for some newly generated images, thereby affecting the results. To address this, the authors propose a simple yet effective approach to assess the reliability of supervisory loss by estimating the uncertainty in the generated images. Specifically,  during the training process they measure the variability between two different images generated under the same condition as an indicator of the uncertainty in the supervisory loss, suggesting that a high variability implies an unreliable signal. This method automatically corrects the supervisory signals, thereby enhancing the quality of the generated models. The conclusions are supported by detailed experimental evidence.

**Strengths:**

The method is straightforward and effective, and the paper is well-organized and easy to understand.

**Weaknesses:**

1. There is a lack of important comparisons, particularly the absence of direct comparison with methods that use auxiliary networks to measure uncertainty (e.g.  Learning model uncertainty as
variance-minimizing instance weights(ICLR 24), It can be used to conduct an ablation experiment to compare who is the better way to measure uncertainty for Conditional Image Generation), discussed on line 214, which is a significant oversight.

2. The motivation section in the introduction is somewhat abrupt and some statements, such as those mentioned on line 74 regarding "the denoised image at t = 0 aligns well with the input condition," are not intuitive and require clearer explanation.

**Questions:**

see Weaknesses

---

> ### Author Response · Authors · 2024-11-22
> **Response for Reviewer gccp**
>
> **Comment:**
>
> Dear Reviewer gccp
>
> Thank you for your appreciation and constructive feedback on our manuscript. We are confident that your insights will greatly improve the writing quality and experimental integrity of our work. We have addressed several writing issues raised by you and other reviewers. Additionally, we conducted ablation experiments to explore more effective uncertainty measurements in conditional generation.
>
>
> **Weaknesses:**
>
> (1) Conduct an ablation study on uncertainty measurement for conditional image generation.
>
> Thank you for your precious suggestion! We would like to clarify that, besides the overfitting problem discussed on line 214, REVAR [1] is an instance-level approach with an auxiliary network to capture predictive uncertainty. This approach is not well-suited for pixel-level conditional image generation. In contrast, our method operates at the pixel level, providing greater precision.
>
> As suggested, we run experiments following REVAR [1], designing an auxiliary network $g(x)$ to directly regress uncertainty. Specifically, the reward model with $g(x)$ is trained using cross-entropy loss on labeled examples from ADE20K, and then leveraged to refine reward optimization. Experimental comparisons show that our uncertainty measurement is more competitive in conditional generation.
>
> | Method             | mIoU | FID    | CLIP  |
> |:---:|:---:|:---:|:---:|
> |Controlnet++ |43.64 |29.49 |31.96|
> |REVAR |42.15 |31.27 |30.97|
> |Ctrl-U | 46.49 |28.61 |31.26|
>
> (2) The motivation section in the introduction should more clearly explain that "the denoised image at $t=0$ aligns well with the input condition" for motivation.
>
> We apologize if the intuitions were not made clear. $t=0$ denotes that we do not add any Gaussian noise during the diffusion process to generation the image. The denoised image at $t=0$ is different from the input image, due to compression process in the diffusion models. We observe that the denoised image at $t=0$ also suffers inaccurate feedback from the reward model.
> We have incorporated the following illustrations into the paper with blue color.
>
> During the diffusion training, we add different levels of Guassian noise to the input, which increases the reconstruction challenge. It also leads to the distribution discrepancy between the generated image and the real image. As the timestep $t$ increases, such generation discrepancy increases. The reward model has not "seen" such generation discrepancy before, resulting false segmentation prediction, even if the generation is correctly aligned with the condition.
> Due to the diffusion compression process, we observe that even if $t=0$, there is some  mis-alignment feedback as well.
>
> **References:**
>
> [1] Nishant Jain, Karthikeyan Shanmugam, and Pradeep Shenoy. Learning model uncertainty as variance-minimizing instance weights. International Conference on Learning Representations (ICLR), 2024.

---

> ### Author Response · Authors · 2024-11-22
> **Sincere Request for Further Discussions**
>
> Dear Reviewer gccp,
>
> Thanks again for your great efforts and constructive advice in reviewing this paper! With the discussion period drawing to a close, we expect your feedback and thoughts on our reply. We put a significant effort into our response, with several new experiments and discussions. We sincerely hope you can consider our reply in your assessment. We look forward to hearing from you, and we can further address unclear explanations and remaining concerns if any.
>
> Regards,
>
> Authors

---

### Official Review · Reviewer_QrcQ · 2024-11-10

**Soundness:** 3
**Presentation:** 3
**Contribution:** 3
**Rating:** 6
**Confidence:** 2

**Summary:**

The authors investigate the reward model used in conditional generation diffusion models and find that it tends to output wrong conditions (e.g., segmentation masks) on the de-noised images. They propose leveraging the output uncertainty on two different generated results from different denoising timestamps. They reduce the loss weight of pixels with high uncertainty while increasing ones with low uncertainty.

**Strengths:**

1. The proposed method is intuitive and easy to incorporate.
2. The quantitative results outperform the baselines by a large margin (although some metrics used seem to be questionable, e.g., mIoU from another segmentation model; see Weaknesses)

**Weaknesses:**

1. If I understand correctly, the authors leverage another more powerful segmentation model to evaluate mIoU. However, this seems strange to me. For instance, in Fig. 3, while the authors circled out the additional green masks by ControlNet++, I think this shouldn't be regarded as an error. Because the green boxes seem to be windows, and the powerful segmentation model predicts these masks, it could be that ControlNet++ did a great job generating the windows based on the concept of the pinkish building mask.
2. Please see Questions.

**Questions:**

1. The annotation in Fig. 2 and Eq. 2 needs to be clarified. Specifically, is the KL divergence computed between the input control (c) and c1, c2 separately? Or is it computed only between c1 and c2? While Fig. 2 denotes the former, Eq. 2 denotes the latter.
2. What is the range of t1 and t2? Also, do the optimal t1 and t2 vary on different datasets/tasks?
3. Visualization of inaccurate prediction from the reward model and the corresponding uncertainty prediction. Are they aligned? In other words, does the uncertainty reduce the weight loss of positions where the predicted condition matches the input but somehow has a significant loss (the reward model predicts the wrong segmentation mask)?
4. Line 360, can the authors elaborate more on "After communicating with the authors, we re-implemented the CLIP-Score for ControlNet++ (Li et al., 2024b) and marked it with ∗ in Table 3"?
5. Does the reward model impact the performance if a different one is used? Also, does the consistency prediction generalize well to different distributions? (e.g., different segmentation datasets)

---

> ### Author Response · Authors · 2024-11-22
> **Response for Reviewer QrcQ (Part #1)**
>
> Dear Reviewer QrcQ
>
> We appreciate QrcQ's thorough review and valuable suggestions. We believe the proposed comments, including clarifying annotations in the method section and improving the presentation of figures and results, will greatly enhance the clarity and quality of our paper. We have submitted a revised version that addresses these concerns in more detail.
>
> **Weaknesses:**
>
> (1) Why choose a more powerful segmentation model for evaluation?
>
> Thank you for raising the concern about using a more powerful segmentation model for evaluation. Our experimental setup follows the previous work, i.e., ControlNet++ [1], which utilizes another segmentation model to compute mIoU. This design choice is intended to prevent reward hacking [2, 3], ensuring that performance improvements are not a result of overfitting to the reward model's biases.
>
> (2) Fig. 3: Does ControlNet++ generate plausible results based on the segmentation mask?
>
> Thanks for your detailed feedback in Fig. 3. We apologize for any confusion caused by the inadequate label descriptions and will include detailed category illustrations in the revised version. **Notably, the green boxes you identified are labels for awnings, not windows.** The input condition specified for the indicated position is building, yet ControlNet++ generated awnings, failing to align with the given condition. In contrast, the image generated by Ctrl-U shows more accurate alignment with the input control, validating the effectiveness of our methodology in improving the controllability.

---

> ### Author Response · Authors · 2024-11-22
> **Response for Reviewer QrcQ (Part #2)**
>
> **Questions:**
>
> (1) The annotation in Eq. 2 and Fig. 2.
>
> Thank you for pointing out this issue. In our method, we calculate the KL divergence between $\hat{c}_1$ and $\hat{c}_2$, i.e., $U_1, U_2$ in Eq. 2 and Fig. 2. Subsequently, we adopt a condition alignment loss between the input condition $c$ and $\hat{c}_1, \hat{c}_2$, respectively, i.e., $\mathcal{L}_1^c$, $\mathcal{L}_2^c$ in Eq. 3,
> while incorporating our estimated uncertainty to adaptively optimize the reward training process, i.e., $\mathcal{L}_1^u$, $\mathcal{L}_2^u$ in Eq. 4.
> This is why the orange arrows in Fig. 2 point from the input control $c$ to $\mathcal{L}_1^u$ and $\mathcal{L}_2^u$, respectively. For clarity, we have updated a more detailed explanation in the caption of Fig. 2.
>
> (2) What is the range of $t_1$ and $t_2$, and do the optimal values vary across different datasets or tasks?
>
> We have conducted ablation studies for $t_1, t_2$ on the segmentation task of the ADE20K dataset. As shown in Table 5 (a) and (b), when both $t_1<400$ and $t_2<400$, and $|t_1-t_2|$ is small, i.e., 1 and 3, a balance between controllability and image quality is achieved. When $|t_1 - t_2| = 0$, the reward discrepancy between the generated images is small, only stemming from resampled noise $\epsilon$, thus failing to serve as a better uncertainty indicator.
>
> Thank you for the suggestion regarding whether optimal $t_1$ and $t_2$ vary across different datasets or tasks. Consequently, we conducted ablation studies on other dataset (COCO-Stuff) and task (depth), and find that the optimal $t_1$ and $t_2$ are not sensitive to dataset and task. In most cases,  $|t_1-t_2|=1$ or 3 is relatively good. Controllability and image quality studies under different conditional controls and datasets below:
>
>
> | Method             | Segmentation (ADE20K) | | | Segmentation (COCO-Stuff) | |  | Depth (MultiGen-20M) | | |
> |:---:|:---:|:---:|:---:| :---:|:---:|:---:| :---:|:---:|:---:|
> |                     | mIoU | FID    | CLIP  | mIoU  |  FID   |   CLIP | RMSE   | FID   | CLIP |
> |ControNet++| 43.64 | 29.49 | 31.24 | 34.56 | 19.29 |  30.93 |  28.32 |16.66 |**32.02** |
> | \|t1 − t2\| = 0 |45.33 |29.01  |30.92  | 45.36 |15.94 | 31.19| 26.93 |16.26 |31.67|
> | \|t1 − t2\| = 1 |46.49 | **28.61** | **31.26** | **49.91** |**15.79** | 31.23| 25.86 |**15.48** |31.72|
> | \|t1 − t2\| = 3| **46.72**| 29.01 |31.11  |48.01  |15.99 | 31.11 |**24.63**  |15.53 |31.17|
> |\|t1 − t2\| = 5 | 45.94| 29.21 |31.16 |47.74 |15.95 |**31.26**   |26.59  |15.93  |31.64|
> |\|t1 − t2\| = 9 | 44.88| 28.94 |31.19 |47.67 |15.96 | 31.19  |25.40  |15.79  |31.58|

---

> ### Author Response · Authors · 2024-11-22
> **Response for Reviewer QrcQ (Part #3)**
>
> (3) Do the inaccurate predictions align with high uncertainty areas? Will the uncertainty harm accurate areas?
>
> Thank you for the great question. We observe that inaccurate rewards and estimated uncertainty are highly correlated, i.e., the areas with incorrect rewards exhibit higher uncertainty. For further details, please refer to the newly added Appendix A.2. Therefore, the uncertainty mainly reduces the weight loss of inaccurate areas, and has few impacts on accurate areas.
>
> (4) The unclear explanation of CLIP-Score for ControlNet++ in Table 3.
>
> We apologize for the confusion. We contacted the authors of ControlNet++ and discovered that they made an error when reporting the data. Specifically, we were unable to replicate the CLIP-Score reported in ControlNet++ [1] using official checkpoints and evaluation code from the official repository.
> Therefore, we have included our re-evaluated CLIP-Score for ControlNet++ in Table 3, marked with $^*$ and gray.
>
> (5) The impact of different reward models on performance and the generalization of consistency predictions across different segmentation datasets.
>
> This is an intriguing point. We conduct additional experiments and observe that the choice of reward models, e.g., UperNet-R50, DeepLabv3-MBv2 and FCN-R101, impacts the final performance. Notably, the proposed uncertainty-aware reward modeling is scalable to different reward models to further enhance both controllability and image quality.
> Comparative results on ADE20K dataset below (using SD1.5 as the backbone).
>
>
> | Method             | UperNet-R50 | | | DeepLabv3-MBv2 | |  | FCN-R101 | | |
> |:---:|:---:|:---:|:---:| :---:|:---:|:---:| :---:|:---:|:---:|
> |                     | mIoU | FID    | CLIP  | mIoU  |  FID   |   CLIP | mIoU   | FID   | CLIP |
> |w/o uncertainty| 43.64 | 29.49 | 31.24 | 45.03 | 29.75 | **31.01** | 46.58 | 29.52 | 30.92|
> |w/ uncertainty |**46.49** | **28.61** | **31.26** | **46.88** | **28.52** | 30.94 | **48.23** | **28.84** | **31.00**|
>
> Furthermore, we would like to clarify that we have validated the generalization of consistency predictions across different segmentation datasets, i.e., ADE20K and COCO-Stuff, in Tables 1-3.
>
> **References:**
>
> [1] Ming Li, Taojiannan Yang, Huafeng Kuang, Jie Wu, Zhaoning Wang, Xuefeng Xiao, and Chen Chen. Controlnet++: Improving conditional controls with efficient consistency feedback. European Conference on Computer Vision (ECCV), 2024.
>
> [2] Ahmed M Ahmed, Rafael Rafailov, Stepan Sharkov, Xuechen Li, and Sanmi Koyejo. Scalable ensembling for mitigating reward overoptimisation. International Conference on Learning Representations (ICLR), 2024.
>
> [3] Jacob Eisenstein, Chirag Nagpal, Alekh Agarwal, Ahmad Beirami, Alex D’Amour, DJ Dvijotham, Adam Fisch, Katherine Heller, Stephen Pfohl, Deepak Ramachandran, et al. Helping or herding? reward model ensembles mitigate but do not eliminate reward hacking. Conference on Language Modeling (CoLM), 2024.

---

> ### Author Response · Authors · 2024-11-22
> **Sincere Request for Further Discussions**
>
> Dear Reviewer QrcQ,
>
> Thanks again for your great efforts and constructive advice in reviewing this paper! With the discussion period drawing to a close, we expect your feedback and thoughts on our reply. We put a significant effort into our response, with several new experiments and discussions. We sincerely hope you can consider our reply in your assessment. We look forward to hearing from you, and we can further address unclear explanations and remaining concerns if any.
>
> Regards,
>
> Authors

---

> > ### Comment · Reviewer_QrcQ · 2024-12-02
> > **Thanks to the author for detailed responses**
> >
> > I've reviewed the authors' responses, which clarify most of my concerns. I will maintain my initial rating.

---

### Meta-Review · Area_Chair_jt4b · 2024-12-20

**Metareview:**

Summary:

This paper identifies an issue in the perceptual loss used in diffusion-based conditional image generation, where incorrect supervisory signals can degrade quality. Correspondingly, the authors propose an uncertainty-aware reward modeling method (Ctrl-U) that uses prediction variability between two generated images from different denoising timestamps as an uncertainty indicator to adaptively adjust loss weights – rewards with lower uncertainty are given higher weights, while those with higher uncertainty are down-weighted. This approach demonstrates improved controllability, reliability, and state-of-the-art conditional image generation performance.

Strengths:

Based on the reviewers' comments, the proposed method is intuitive, sound, and effectively addresses the critical issue of uncertainty in reward models for conditional image generation. It outperforms baselines by a significant margin in quantitative results and avoids the need for additional modules to assess reward uncertainty, making it a straightforward and effective solution. The paper is well-organized and easy to understand.

Weaknesses:

Based on the reviewers' comments, there was a perceived lack of technical novelty, as the method simply generates two images, calculates uncertainty, and reweights loss, which some reviewers (Reviewer 4HGt) felt was trivial and computationally expensive, suggesting that an uncertainty predictor could provide a more efficient solution and be used for an ablation study (Reviewers 4HGt and gccp). Concerns were also raised about the trade-off between controllability/alignment and image quality. Additionally, the fairness of the comparisons was questioned, along with the increased computational costs. Some aspects of the evaluation metrics and methodology configuration, as well as their impact, were also unclear.

Discussion:

During the author-reviewer discussion phase, the authors provided detailed clarifications and additional experimental results. Reviewers QrcQ and 4HGt acknowledged that the rebuttal effectively addressed their concerns. Although Reviewers gccp and xjaQ did not respond to the rebuttal, their initial ratings were borderline accept. By the end of the discussion phase, all reviewers rated the paper positively.

AC Recommendation:

AC reads the authors' rebuttal and found that it effectively addressed the concerns raised by Reviewers gccp and xjaQ. There is no basis to overturn the reviews, and AC agrees with the recommendation. In the camera-ready version, the authors should include the additional experiments and explanations provided in the discussion.

**Additional Comments On Reviewer Discussion:**

During the author-reviewer discussion phase, the authors provided detailed clarifications and additional experimental results. Reviewers QrcQ and 4HGt acknowledged that the rebuttal effectively addressed their concerns. Although Reviewers gccp and xjaQ did not respond to the rebuttal, their initial ratings were borderline accept. By the end of the discussion phase, all reviewers rated the paper positively.

AC reads the authors' rebuttal and found that it effectively addressed the concerns raised by Reviewers gccp and xjaQ. There is no basis to overturn the reviews, and AC agrees with the recommendation. In the camera-ready version, the authors should include the additional experiments and explanations provided in the discussion.

---

### Decision · Program_Chairs · 2025-01-22

Accept (Poster)